# The Effect of Targeted Hyperoxemia on Brain Immunohistochemistry after Long-Term, Resuscitated Porcine Acute Subdural Hematoma and Hemorrhagic Shock

**DOI:** 10.3390/ijms25126574

**Published:** 2024-06-14

**Authors:** Franziska Münz, Thomas Datzmann, Andrea Hoffmann, Michael Gröger, René Mathieu, Simon Mayer, Fabian Zink, Holger Gässler, Eva-Maria Wolfschmitt, Melanie Hogg, Enrico Calzia, Pierre Asfar, Peter Radermacher, Thomas Kapapa, Tamara Merz

**Affiliations:** 1Institute for Anesthesiological Pathophysiology and Process Engineering, Ulm University, 89081 Ulm, Germanypeter.radermacher@uni-ulm.de (P.R.); 2Department of Anesthesiology and Intensive Care Medicine, University Hospital Ulm, 89081 Ulm, Germany; 3Department of Neurosurgery, German Federal Armed Forces Hospital Ulm, 89081 Ulm, Germany; 4Department of Anesthesiology, Intensive Care Medicine, Emergency Medicine and Pain Therapy, German Armed Forces Hospital Ulm, 89081 Ulm, Germany; 5Department of Intensive Care and Hyperbaric Medicine, University Hospital Angers, 49045 Angers, France; 6Department of Neurosurgery, University Hospital Ulm, 89081 Ulm, Germany

**Keywords:** traumatic brain injury, blood–brain barrier, oxidative/nitrosative stress, cystathionine-β-synthase, cystathionine-γ-lyase, oxytocin, oxytocin receptor

## Abstract

Epidemiological data suggest that moderate hyperoxemia may be associated with an improved outcome after traumatic brain injury. In a prospective, randomized investigation of long-term, resuscitated acute subdural hematoma plus hemorrhagic shock (ASDH + HS) in 14 adult, human-sized pigs, targeted hyperoxemia (200 < P_a_O_2_ < 250 mmHg vs. normoxemia 80 < P_a_O_2_ < 120 mmHg) coincided with improved neurological function. Since brain perfusion, oxygenation and metabolism did not differ, this post hoc study analyzed the available material for the effects of targeted hyperoxemia on cerebral tissue markers of oxidative/nitrosative stress (nitrotyrosine expression), blood–brain barrier integrity (extravascular albumin accumulation) and fluid homeostasis (oxytocin, its receptor and the H_2_S-producing enzymes cystathionine-β-synthase and cystathionine-γ-lyase). After 2 h of ASDH + HS (0.1 mL/kgBW autologous blood injected into the subdural space and passive removal of 30% of the blood volume), animals were resuscitated for up to 53 h by re-transfusion of shed blood, noradrenaline infusion to maintain cerebral perfusion pressure at baseline levels and hyper-/normoxemia during the first 24 h. Immediate postmortem, bi-hemispheric (i.e., blood-injected and contra-lateral) prefrontal cortex specimens from the base of the sulci underwent immunohistochemistry (% positive tissue staining) analysis of oxidative/nitrosative stress, blood–brain barrier integrity and fluid homeostasis. None of these tissue markers explained any differences in hyperoxemia-related neurological function. Likewise, hyperoxemia exerted no deleterious effects.

## 1. Introduction

The post-traumatic mortality and morbidity are contingent upon the presence or absence of traumatic brain injury (TBI) and/or hemorrhagic shock (HS) [1]. Throughout HS treatment, supplementary O_2_ is often administered to enhance the amount of physically dissolved O_2_, aiming to expedite the restoration of tissue O_2_ levels during impairment of O_2_ transport caused by blood loss [2].

Hyperoxemia, however, may exert potentially adverse effects, including elevated generation of reactive oxygen species (ROS), in particular during ischemia/reperfusion sequences, e.g., restoration of blood flow during resuscitation from HS [2]. Nonetheless, after TBI, epidemiological data showed that the presence of moderate hyperoxemia (150 < P_a_O_2_ < 200 mmHg during the initial 24 h of intensive care unit (ICU) stay) coincided with improved functional outcomes and survival [3]. In contrast, more recent data from the Center–TBI cohort found that long-term morbidity and mortality were aggravated with every 10 mmHg increase in supranormal arterial O_2_ partial pressure (P_a_O_2_) [4]. Finally, in a long-term, resuscitated porcine model of acute subdural hematoma (ASDH) and HS in adult and cardiovascular healthy animals, hyperoxemia targeting P_a_O_2_ of 200–250 mmHg was associated with improved neurological function when compared to standard treatment (P_a_O_2_ 80–120 mmHg) [5]. In these experiments, neither parameters of cerebral tissue perfusion (intracranial and cerebral perfusion pressure) and oxygenation (brain tissue O_2_ partial pressure) nor metabolism (tissue microdialysis for glucose, lactate, pyruvate, and glutamate) showed any differences between the two experimental groups. Hence, the mechanisms allowing us to explain any effects of this targeted hyperoxemia on neurological function remain largely underexplored.

In a long-term, resuscitated porcine model, we had previously shown that ASDH alone, i.e., without additional HS, was associated both with markedly higher extravascular albumin accumulation as a marker of blood–brain barrier dysfunction, as well as nitrotyrosine formation as a marker of oxidative and nitrosative stress in the blood-injected brain hemisphere when compared to the contralateral regions, which had only undergone neurosurgical instrumentation [6]. Hippocampal nitrotyrosine staining was also significantly increased in hyperoxia-exposed rats after controlled cortical impact-induced TBI [7]. However, delayed analysis of neuronal survival in that study did not show any significant difference between the hyperoxic and normoxic groups. In mice that had undergone combined TBI and HS, hyperoxia was also associated with signs of aggravated tissue oxidative stress, as demonstrated by increased ascorbate depletion, but, in contrast, hippocampal neuron survival improved [8].

Both oxytocin (OT) as well as its receptor (OT-R) and the hydrogen sulfide (H_2_S) system are believed to be of pivotal importance in the stress response to physical and psychological trauma, but also in fluid homeostasis, e.g., blood loss. In fact, in hemorrhaged rats, OT formation was significantly decreased both in the hypothalamus and in the neurohypophysis [9]. Moreover, tissue OT was shown to be significantly decreased in the ipsilateral striatum in a mouse model of intracerebral hemorrhage, whereas the OT-R expression showed the opposite response, i.e., a significant increase [10]. Our immuno-histochemical characterization of the H_2_S and OT systems in brain specimen from swine that had undergone ASDH alone without HS showed expression of the OT-R on both sides, i.e., in the brain hemisphere affected by the acute subdural hematoma (ipsilateral) as well as in the control (contralateral) hemisphere. OT, however, was only expressed on the ipsilateral side. Similarly, cystathionine-β-synthase (CBS), one of the two major H_2_S-producing enzymes, was also predominantly detected on the ipsilateral side. In contrast, the staining for cystathionine-γ-lyase (CSE), the other of the two major H_2_S-producing enzymes, was primarily present in the contralateral hemisphere, i.e., that without subdural blood injection [11]. Finally, after murine TBI, the expression of CBS was significantly decreased in the ipsilateral hemisphere, but reached basal level after 7 days [12]. It is noteworthy that the aforementioned rodent studies were conducted without hemodynamic support to maintain cerebral perfusion pressure (CPP), which is crucial for the outcome of TBI patients. Moreover, animals breathed pure O_2_ rather than an inspiratory O_2_ concentration titrated to achieve pre-defined P_a_O_2_ targets; in other words, they were exposed to potentially excessive hyperoxia.

Hence, the effect of targeted hyperoxemia after combined ASDH-induced TBI and HS on brain immunohistochemistry is still unknown. Therefore, this post hoc study of brain tissue specimens available as part of the above-mentioned interventional study [5] investigated the effect of targeted hyperoxemia on cerebral tissue markers of oxidative/nitrosative stress, blood–brain barrier integrity and fluid homeostasis.

## 2. Results

Figure 1 and Figure 2 show representative images (left side) and the quantitative analysis (right panels) of the cerebral tissue detection of albumin (Figure 1) as, when occurring outside of the vasculature, a marker of blood–brain barrier integrity in both gray (upper graphs) and white matter (lower graphs) as well as for nitrotyrosine (Figure 2) as a marker of oxidative and nitrosative stress. Neither nitrotyrosine expression nor extravascular albumin accumulation showed any significant intergroup difference, regardless of the presence/absence of ASDH or of hyperoxemia. While a substantial amount of albumin was detected in the gray matter and a noteworthy portion of tissue in the white matter, indicating compromised blood–brain barrier function, nearly no nitrotyrosine was present in the gray matter, and very little in the white matter.

Figure 3 and Figure 4 show representative images (left panels) and the quantitative analysis (right panels) of the cerebral tissue staining for oxytocin (Figure 3) and oxytocin receptor (Figure 4) in both gray (top graphs) and white matter (bottom graphs). Neither oxytocin nor the expression of oxytocin receptor exhibited any significant intergroup variance, irrespective of the presence or absence of ASDH or hyperoxemia.

Representative images (left panels) and the quantitative analysis (right panels) of the cerebral tissue staining for the H_2_S-producing enzymes CBS and CSE are presented in Figure 5 and Figure 6 (gray matter: upper graphs; white matter: lower graphs). Neither CBS (Figure 5) nor CSE (Figure 6) showed any significant intergroup differences.

## 3. Discussion

The aim of this investigation was to assess the protein levels of extravascular albumin, nitrotyrosine, CBS, CSE, OT, and OT-R in the prefrontal cortex following ASDH and HS, and subsequent resuscitation with targeted hyperoxemia vs. standard treatment. The main findings of the present study were (i) no significant differences in expression of any of these parameters, (ii) without major effect at all of hyperoxemia.

In our study, the presence of albumin extravasation served as an indication of the destruction of the blood–brain barrier, which is consistent with findings from a previous study conducted on gerbils, wherein immunohistochemical analysis demonstrated increased permeability of the blood–brain barrier to serum albumin after infarction in the hippocampus [13]. In comparison to our prior investigations in swine, the following disparities were observed: following HS alone without simultaneous ASDH or any other direct acute brain injury, no alterations in the levels of extravascular albumin had been noted [14], whereas following ASDH alone, i.e., direct acute brain injury, but without concomitant circulatory depression, a significant increase in extravascular albumin was detected in the ipsilateral, i.e., blood-injected brain hemisphere [6]. In the present study, the concurrent occurrence of ASDH and HS induced an extravasation of albumin. Overall, these findings suggest that only the combined presence of acute brain injury and circulatory (in other words, simultaneous regional (cerebral) and systemic (circulatory shock)) challenge is mandatory to produce disruption of the blood–brain barrier in the contralateral hemisphere, i.e., that without blood injection, ultimately resulting in albumin extravasation.

The present study revealed only a little nitrotyrosine expression, regardless of the blood-injected vs. instrumented-only hemisphere. In comparison, in the previously mentioned study focusing on HS alone, no expression of nitrotyrosine was detected [14]. Conversely, following ASDH alone, a significantly higher level of nitrotyrosine could be observed in the ipsilateral hemisphere [6]. This might suggest that ASDH alone per se may induce ipsilateral oxidative stress. It should be noted, however, that in that study, the volume of blood injected into the subdural was markedly higher, i.e., 20 mL corresponding to approximately 12.5% of the brain volume, than in the present study, i.e., 0.1 mL/kgBW ≈ 7–8 mL in animals with slightly higher body weight (median 75 [5] vs. 65 [6] kg). This approach was chosen to keep the volume of blood injected into the subdural below 10% of the intracranial volume and, therefore, to avoid major intracranial hypertension with a subsequent fall of CPP during the HS phase with potentially irreversible brain injury.

Targeted hyperoxemia neither influenced the levels of nitrotyrosine (a marker of oxidative and nitrosative stress) in the ipsilateral nor in the contralateral hemisphere. In both experimental animals and patients, the available data on the impact of hyperoxemia on oxidative stress following TBI is inconclusive, with reports of exacerbated [7], unchanged [15,16], or even diminished [17] markers of radical damage. Even prolonged hyperoxemia, comparable to our investigation, for up to 14 days did not impact serum markers related to oxidative stress, inflammation, or neurological injury [16]. Moreover, it is worth noting that most of the aforementioned studies focused on short-term (up to 4 h) pure O_2_ ventilation, resulting in P_a_O_2_ levels between 300 and 450 mmHg, rather than moderate hyperoxemia within a well-defined target range, as in our experiment. Thus, the results from that study align with our own findings, indicating that moderate hyperoxia within the chosen limits may be safe with respect to serum or cerebral tissue markers of oxidative stress.

Targeted hyperoxemia suppressed the elevation of oxytocin in the ipsilateral brain hemisphere. Moreover, there was reduced expression of oxytocin receptor in the same hemisphere. To date, to the best of our knowledge there are no available data in the literature regarding the impact of hyperoxemia on the cerebral protein levels of oxytocin and oxytocin receptor. In contrast to our findings, divergent expression of oxytocin/oxytocin receptor following intracerebral hemorrhage in a mouse model has been described previously [10]. In rats undergoing subarachnoid hemorrhage by blood injection into the great cistern, oxytocin release sharply increased in the hypothamalic supraoptic nucleus, while no effect was observed in the paraventricular nucleus [18]. It should be underscored, however, that any direct comparison between these rodents and our porcine models is difficult: in addition to the fundamental difference of the rodent, lissencephalic macroscopic brain anatomy when compared to the gyrencephalic porcine brain, in the aforementioned studies animals did not receive any ICU care after blood injection, i.e., no mechanical ventilation (e.g., to control for effects of anesthesia on respiratory drive and, thus, on arterial PCO_2_-related changes in intracranial pressure (ICP)) and/or circulatory support to maintain CPP. Moreover, the breathing gas consisted of pure O_2_ during the instrumentation and blood injection and room air during the observation periods, respectively. Hence, possible effects of excess systemic hyperoxemia and/or later hypoxemia would have remained undetectable. Regardless of the differences in the experimental design, the elevation of OT expression in the damaged brain areas may represent a physiological response of the body utilizing the protective effects of oxytocin [19].

While the OT/OT-R system exhibits interaction with the H_2_S system, it is noteworthy that targeted hyperoxemia, as previously delineated, exerts an influence on the expression of OT and its receptor, yet fails to impact the expression of CBS and CSE within the gray and white matter. While the existing data on CBS expression are currently inconclusive, in the ipsilateral hemisphere, reports indicate both increased expression in swine receiving ICU care [11] and decreased expression in mice without ICU care [12]. There is, to the best of our knowledge, no available data regarding the expression of CSE. However, our study noted consistent expression of CBS and CSE in both hemispheres. This disparity may also be attributed to the substantial difference in the injected blood volume inducing ASDH between our present study (0.1 mL/kgBW) and the previous one by Denoix et al. (20 mL). As mentioned above, this approach had been guided by the reasoning that, in prior investigations of porcine ASDH, injection of a blood volume maximally approaching 10% of the intracranial volume represents the threshold for supratentorial volume tolerance [20,21]. In addition, this approach aimed to prevent pronounced intracranial hypertension and a subsequent fall of CPP during the HS phase in order to avoid potentially irreversible brain damage resulting from low CPP. Finally, as outlined above, a major divergence from Zhang et al. [12] pertains to the utilization of swine in contrast to mice.

The pig serves as a highly relevant translational model due to its structural similarity to the human brain. The presence of gyri and sulci (gyrencephalic brain), a proportional white matter to gray matter ratio and the existence of a tentorium cerebelli distinguish it from the rodent brain and reflect human pathophysiology [19,22]. A critical consideration in translational TBI research pertains to the surface anatomy of the brain: the lissencephalic nature of the rodent brain renders it unsuitable for extrapolation to humans with gyrencephalic brains. The morphology of the gyrencephalic pig brain allows for both gray and white matter responses to injury, closely mirroring human physiology [22]. In lissencephalic brains, “maximum mechanical stress” is experienced proximate to the surface and evenly distributed. Conversely, in gyrencephalic brains, maximum stress is redirected from the surface towards the depths at the base of the sulci, which is reminiscent of conditions in humans [23]. This suggests that the pathophysiology of traumatic injuries in lissencephalic rodents might significantly differ from that in gyrencephalic brains in more developed species like pigs, which share anatomical and morphological features not only in the presence of gyri and sulci but also in the distribution of white matter. Furthermore, an additional limitation of rodent models lies in their failure to replicate the ICU interventions akin to those received by TBI patients, notably the targeted surveillance and adjustment of the CPP.

### Limitations of the Study

Clearly, the results of our study are constrained by the relatively short 55 h timeframe for injury and resuscitation, a limitation imposed by practicability and staff availability for continuous ICU care, which is in sharp contrast to the medical reality, as TBI patients typically remain in the intensive care unit for significantly longer periods. Nevertheless, a recent study emphasized that our swine model incorporating ICU care is of a considerably longer duration when compared to other pig trials [22].

Moreover, it could be argued that the prefrontal cortex is not the appropriate region because the hypothalamus would be more sensitive. However, this region was selected to maintain comparability with our prior study using swine with identical genetic background and of comparable age and body weight, which had focused on the consequences of ASDH alone [6]. Indeed, this approach enabled the detection of specific differences in the consequences of ASDH alone vs. combined ASDH and HS. In both studies, the decision to focus on the prefrontal cortex was driven by its role in coordinating a multitude of complex cognitive processes and regulating behavior. In humans, injuries to this brain region can result in impaired cognitive control, social behavior, and emotional regulation [24]. We reasoned that this selection would be of particular relevance, aiming to identify a correlate elucidating any impact on neurological function (Modified Glasgow Coma Scale) subsequent to targeted hyperoxemia therapy.

## 4. Materials and Methods

### 4.1. Animals

Ethical approval was granted by the Federal Authorities (Tuebingen, Germany) for Animal Research (#1316) and the Animal Care Committee of the University of Ulm. All experiments were conducted in compliance with the European Union Directive 2010/63/EU on the protection of animals used for scientific purposes. Fourteen adult pigs (body weight: 75 kg (73; 76), age: 16 months (15; 18), comprising four females and ten castrated males) were investigated. The pigs belonged to the Bretoncelles–Meishan–Willebrand strain, which is characterized by a diminished activity of the von Willebrand factor, thereby emulating the human coagulation system [6], in contrast to the hypercoagulatory state observed in domestic swine strains [25].

### 4.2. Experimental Protocol

This study is a post hoc analysis of brain sections from a model of combined ASDH and HS in adult pigs in good cardiovascular health [5]. Induction of anesthesia and surgical instrumentation, as well as the experimental protocol of induction of ASDH, HS and resuscitation, has been described in detail previously [5,6] and is graphically illustrated once again in the Appendix A. The volume of blood used to induce ASDH was selected based on the following insights: Firstly, previous studies on porcine ASDH have demonstrated that injecting a blood volume of maximally 10% of the intracranial volume represents the threshold for supra-tentorial volume tolerance [20,21]. Secondly, preliminary work from our own research group on pigs, which only received an ASDH without hemorrhagic shock but with an absolute injected blood volume of 20 mL [6], showed a marked increase in ICP within the first 30 min, in individual animals even reaching values of 80–100 mmHg [6]. Moreover, in that study, even at 2 h after ASDH induction, ICP values were still more than twice as high at the baseline levels, while only a moderate increase was observed in the present experiment. Based on the aforementioned findings, in this experiment, induction of ASDH was performed by injection of 0.1 mL/kgBW of autologous blood through the subdural catheter [5]. Therefore, the cranial vault was exposed, and a craniotomy was executed by perforating openings over the left and right parietal cortices, accompanied by a modest incision of the dura. Adhering to the principles of the 3Rs, the manipulation extended to the contralateral hemisphere to obviate the necessity for supplementary sham experiments. Microdialysis catheters for the determination of glutamate, glucose, pyruvate and lactate, as well as multimodal brain monitoring probes for ICP, brain tissue partial O_2_ pressure (P_bt_O_2_) and temperature measurements were introduced bilaterally into the cerebral parenchyma [5,6]. HS was initiated by passive withdrawal of blood targeting 30% of the calculated blood volume. The deceleration of blood extraction was implemented, as required, to uphold CPP, i.e., the difference between mean arterial pressure (MAP) and ICP ≥ 50 mmHg. Subsequent to 2 h of concurrent ASDH and HS, resuscitation was initiated. This encompassed the re-transfusion of shed blood within 30 min, fluid resuscitation and the continuous intravenous administration of noradrenaline, titrated to sustain MAP at pre-shock levels and CPP > 60 mmHg [5]. Albeit already presented in Datzmann et al. [5], the data are shown in the supplementary material upon the reviewer’s request (Appendix A). During the first 24 h of resuscitation, the animals were randomly assigned to the control group receiving normoxemia (P_a_O_2_ 80–120 mmHg) or to the intervention group obtaining targeted hyperoxemia (P_a_O_2_ 200–250 mmHg). To this end, blood gas analyses were conducted hourly during the initial 24 h, and subsequently every 2 h. Any deviations from the target P_a_O_2_ were promptly addressed by adjusting the inspiratory oxygen fraction (F_i_O_2_). After a maximum of 53 h of intensive care treatment or premature cessation of the experiment in accordance with pre-defined termination criteria (i.e., CPP < 60 mmHg despite receiving the maximum vasopressor dose, acute anuric kidney failure and overall survival [5]; at the reviewer’s request, the survival analysis is graphically represented in the Appendix A), pigs were euthanized by intravenous injection of KCl after further deepening of anesthesia. Immediately postmortem, the brain was removed for further tissue analysis.

### 4.3. Immunohistochemistry

Immunohistochemistry was employed to assess cerebral expression levels of extravascular albumin, serving as an indicator of blood–brain barrier integrity, nitrotyrosine as a marker of oxidative and nitrosative stress within the tissue, OT and its receptor, along with the principal H_2_S-producing enzymes CBS and CSE. Immunohistochemistry was selected for several reasons: (i) it is well documented in the literature that densitometric analysis of colorimetric immunohistochemical staining is comparably reliable to Western blotting for quantifying protein levels [26], (ii) significant correlations were observed between densitometric values and those derived from Western blotting analysis [27], and (iii) unlike Western blotting, immunohistochemical evaluation of tissue permits the identification of spatial distribution and protein expression in distinct cell types within the tissue specimen. All brain specimens underwent uniform fixation in a 4% formalin solution for 6 days. Subsequently, each brain was dissected into consecutive coronal sections with a thickness of 4 mm, spanning from the frontal to occipital regions. In instances where the macroscopic section exceeded the dimensions of the embedding cassette (26 × 3 × 4 mm), it was laid flat and further dissected into a maximum of five pieces. This dissection process was conducted in such a way as to facilitate subsequent reconstruction of the complete section. For the purposes of this investigation, the macroscopic section, encompassing the prefrontal cortex, was specifically chosen for detailed analysis. The prefrontal cortex was selected (i) because of its sufficient distance from the ASDH, which primarily affected the parietal cortex, thereby ensuring its macro- and microscopic structure remains intact, (ii) the prefrontal cortex is of great significance in swine, as it is in other mammals, playing a variety of roles in cognitive processing, behavior, and adaptation to the environment, and (iii) to allow for comparison with our previous work in swine that had undergone ASDH alone without HS [6]. The tissue was subsequently dehydrated and encased in paraffin blocks. Sections with a thickness of 3–5 μm underwent de-paraffinization using xylene, followed by rehydration through a graded series of ethanol and deionized water.

Immunohistochemistry was conducted in accordance with the previously outlined methodology [11]. In brief, following de-paraffinization, heat-induced antigen retrieval was performed in citrate solution (pH 6). Subsequently, blocking was achieved using normal goat serum (10%) before incubation with the following primary antibodies: anti-pig Albumin (Abcam, abab79960, RRID:AB_1658916, Cambridge, UK) as a marker of blood–brain barrier dysfunction, H_2_S-producing enzymes anti-CBS (Protein Tech, Planegg-Martinsried, Germany 14787-1-AP, RRID: AB_2070970) and anti-CSE (Protein Tech, Planegg-Martinsried, Germany 12217-1-AP, RRID:AB_2087497), anti-OT (Merck SA, an affiliate of Merck KGaA, Darmstadt, Germany, AB911, RRID:AB_2157629), anti-OT-R (Protein Tech, 2304523045-1-AP, RRID:AB_2827435) and anti-Nitrotyrosine (Merck SA, an affiliate of Merck KGaA, Darmstadt, Germany, ab5411, RRID:AB_177459) as a marker of nitrosative and oxidative stress.

All primary antibodies were adjusted to their optimal dilution based on the manufacturer’s recommendations (Table 1). Negative controls were concurrently conducted [11] by incubating with the diluent instead of the primary antibody to monitor the specificity of the secondary system. Alternatively, when possible, pre-incubation of the primary antibody with the corresponding immunogen peptide was performed. In such cases, brain tissue was incubated with the pre-absorbed primary antibody rather than the regular primary antibody [11]. The primary antibodies were detected utilizing the Dako REAL detection system, which employed alkaline phosphatase-conjugated secondary antibodies (anti-mouse; anti-rabbit). Visualization was achieved using a Fast Red-type chromogen. For each animal, one slide per hemisphere was observed with a Zeiss Axio Imager A1 microscope equipped with a 10× objective. Quantification was performed on 800,000-μm² sections using the Zen Image Analysis Software (Version 3.0, Zeiss, Oberkochen, Germany). The results are reported as the percentage of the positive stained area as a fraction of the total area.

### 4.4. Statistical Analysis

Statistical analysis was conducted using GraphPad Prism Version 8. Inter-group differences were examined utilizing the Kruskal–Wallis test. The data are presented as individual representation with a corresponding median.

## 5. Conclusions

In this post hoc analysis of material from a long-term, resuscitated porcine model with ASDH and HS, the expression levels of cerebral tissue markers associated with oxidative and nitrosative stress, indicators of blood–brain barrier integrity, as well as regulators of fluid homeostasis, did not provide an explanatory framework for the observed variations in neurological function related to targeted hyperoxemia. However, it is noteworthy that targeted hyperoxemia during resuscitation from combined ASDH and HS did not exhibit any apparent adverse effects.

## Figures and Tables

**Figure 1 ijms-25-06574-f001:**
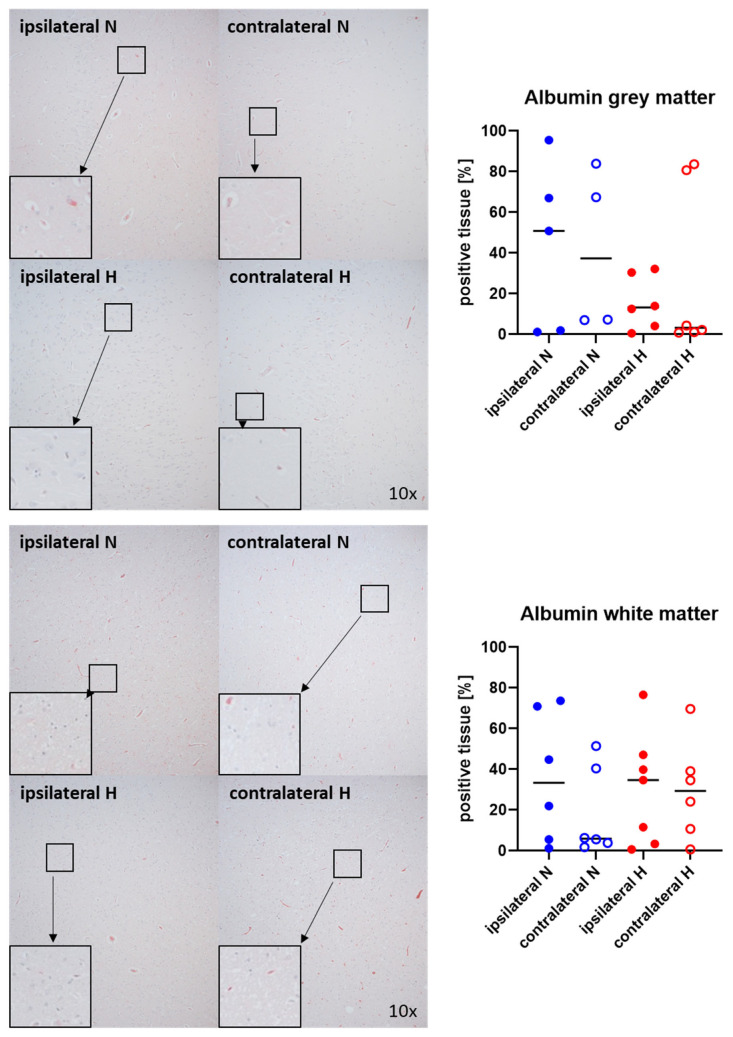
Albumin in the gray matter (**upper panel**; n = 4–6) and white matter (**lower panel**; n = 6–7) of the prefrontal cortex. Representative images (**left panel**), 30× magnified images originating from the black box in the 10× magnified image and quantification of immunohistochemical staining as positive tissue [pink, %] (**right graph**). N: Normoxemia (blue symbols); H: Hyperoxemia (red symbols); ipsilateral: blood-injected hemisphere (solid circles); contralateral: instrumented-only hemisphere (open circles).

**Figure 2 ijms-25-06574-f002:**
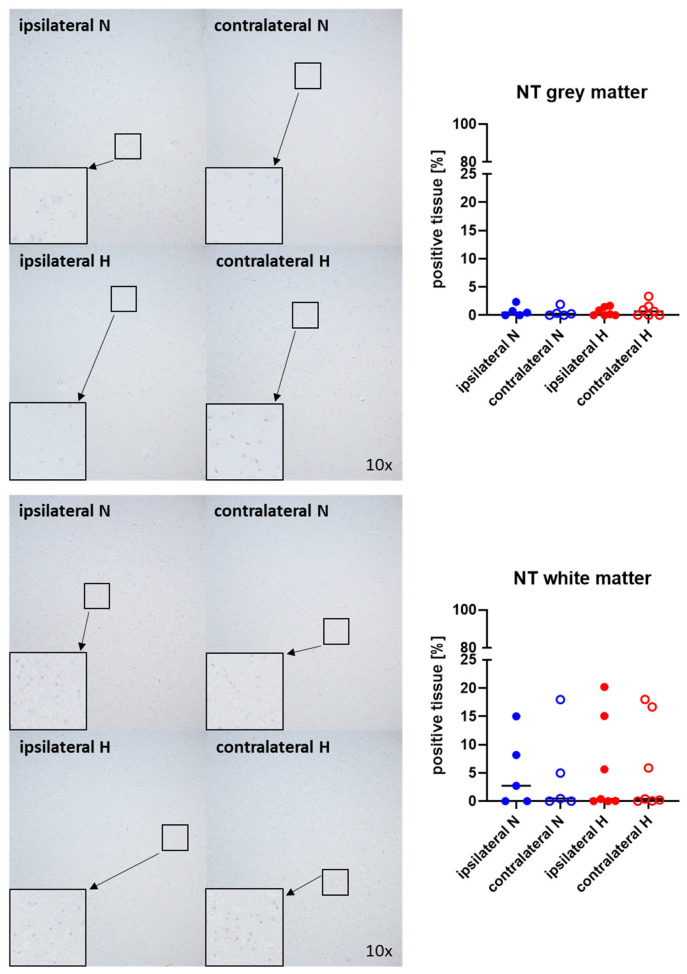
Nitrotyrosine (NT) in the gray matter (**upper panel**; n = 5–7) and white matter (**lower panel**; n = 5–7) of the prefrontal cortex. Representative images (**left panel**), 30× magnified images originating from the black box in the 10× magnified image and quantification of immunohistochemical staining as positive tissue [pink, %] (**right graph**). N: Normoxemia (blue symbols); H: Hyperoxemia (red symbols); ipsilateral: blood-injected hemisphere (solid circles); contralateral: instrumented-only hemisphere (open circles).

**Figure 3 ijms-25-06574-f003:**
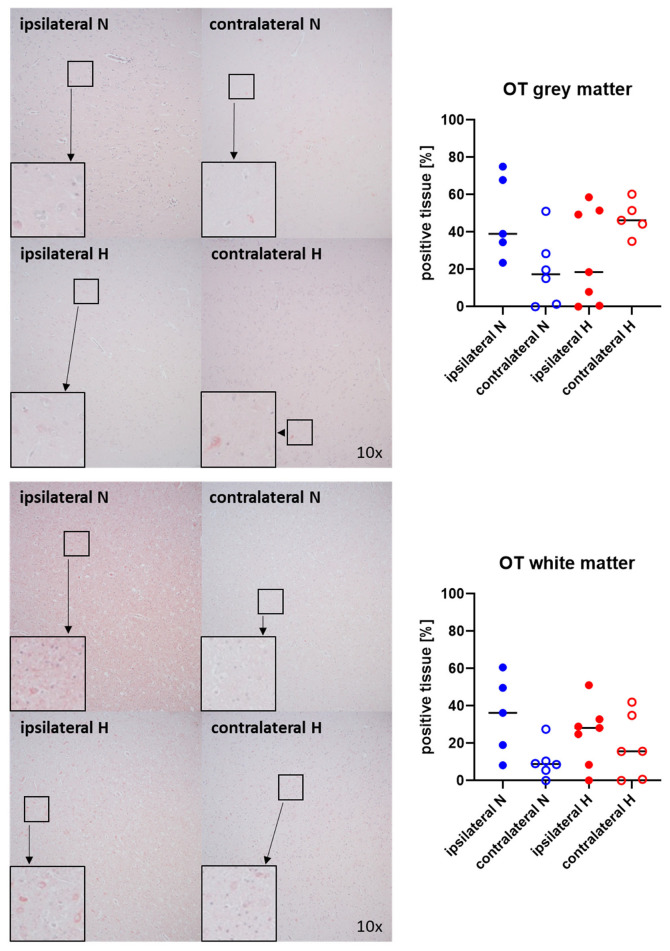
Oxytocin (OT) in the gray matter (**upper panel**; n = 5–7) and white matter (**lower panel**; n = 5–7) of the prefrontal cortex. Representative images (**left panel**), 30× magnified images originating from the black box in the 10× magnified image and quantification of immunohistochemical staining as positive tissue [pink, %] (**right graph**). N: Normoxemia (blue symbols); H: Hyperoxemia (red symbols); ipsilateral: blood-injected hemisphere (solid circles); contralateral: instrumented-only hemisphere (open circles).

**Figure 4 ijms-25-06574-f004:**
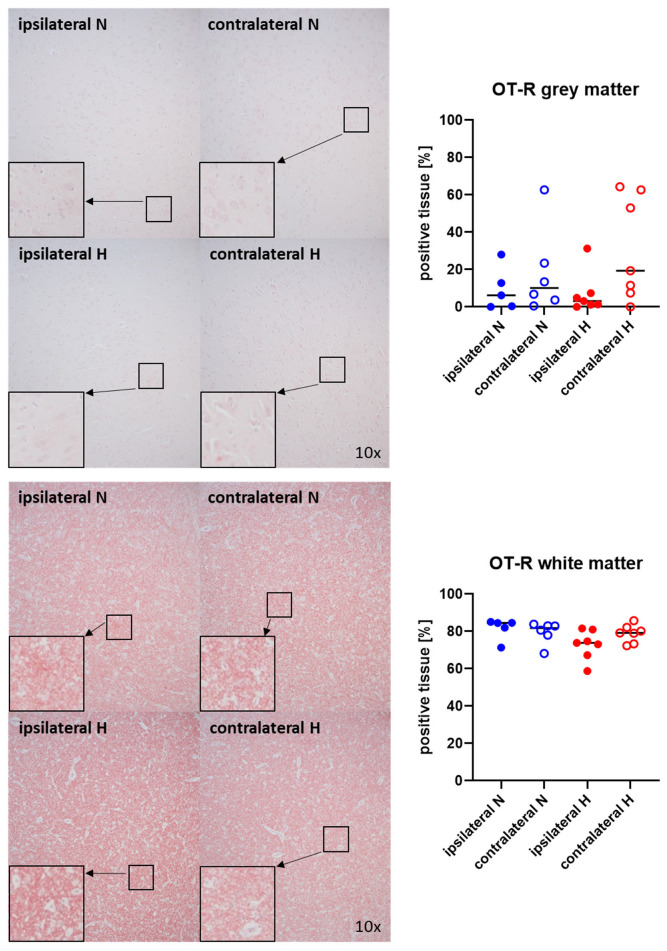
Oxytocin receptor (OT-R) in the gray matter (**upper panel**; n = 5–7) and white matter (**lower panel**; n = 5–7) of the prefrontal cortex. Representative images (**left panel**), 30× magnified images originating from the black box in the 10× magnified image and quantification of immunohistochemical staining as positive tissue [pink, %] (**right graph**). N: Normoxemia (blue symbols); H: Hyperoxemia (red symbols); ipsilateral: blood-injected hemisphere (solid circles); contralateral: instrumented-only hemisphere (open circles).

**Figure 5 ijms-25-06574-f005:**
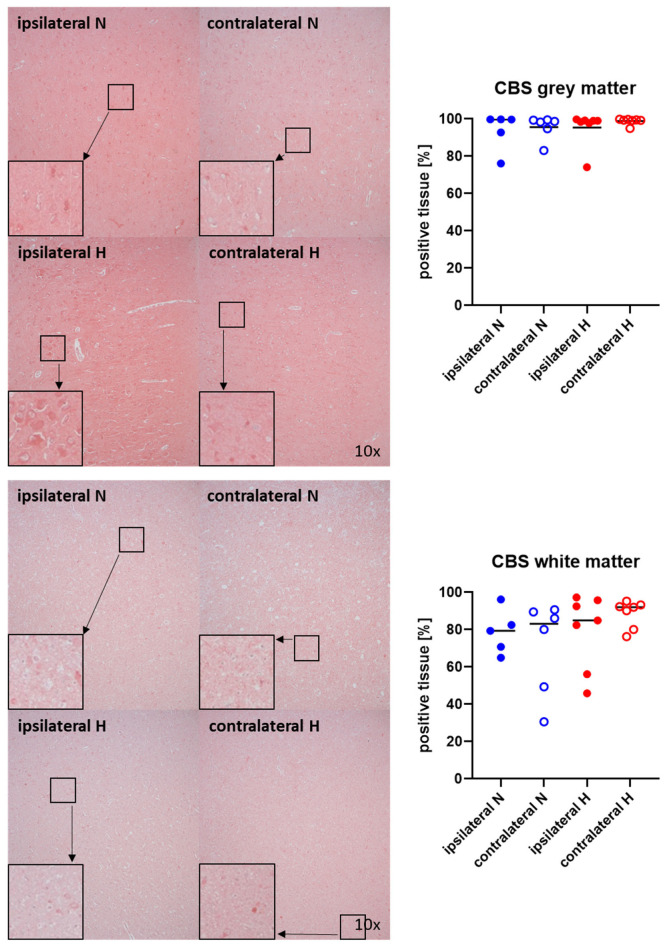
H_2_S-producing enzyme cystathionine-β-synthase (CBS) in the gray matter (**upper panel**; n = 5–7) and white matter (**lower panel**; n = 5–7) of the prefrontal cortex. Representative images (**left panel**), 30× magnified images originating from the black box in the 10× magnified image and quantification of immunohistochemical staining as positive tissue [pink, %] (**right graph**). N: Normoxemia (blue symbols); H: Hyperoxemia (red symbols); ipsilateral: blood-injected hemisphere (solid circles); contralateral: instrumented-only hemisphere (open circles).

**Figure 6 ijms-25-06574-f006:**
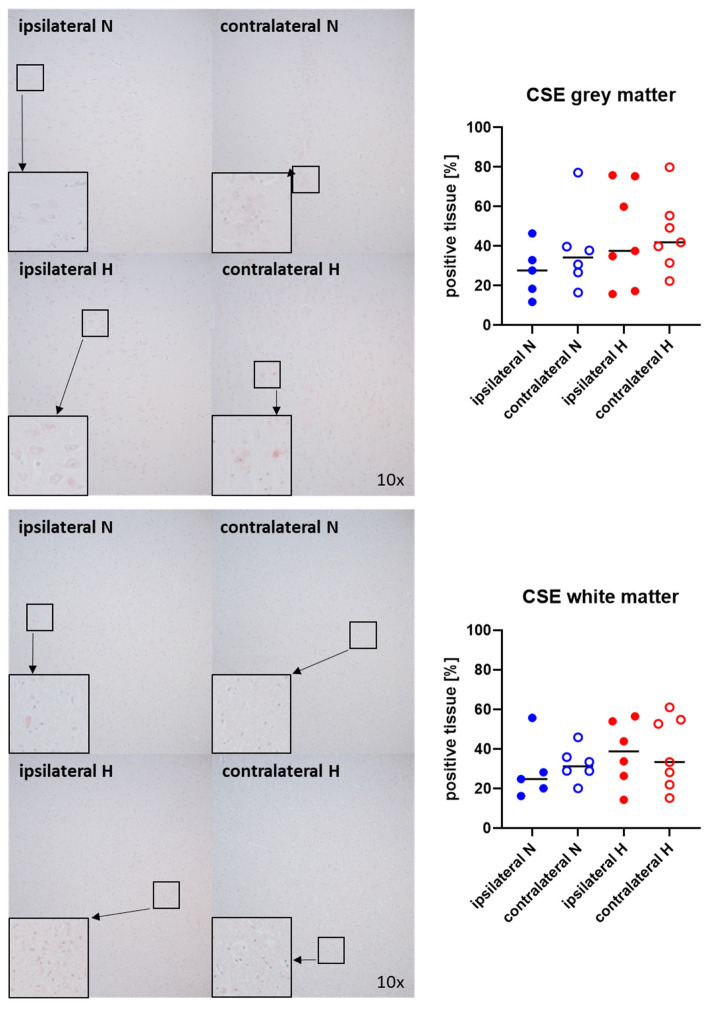
H_2_S-producing enzyme cystathionine-γ-lyase (CSE) in the gray matter (**upper panel**; n = 5–7) and white matter (**lower panel**; n = 5–7) of the prefrontal cortex. Representative images (**left panel**), 30× magnified images originating from the black box in the 10× magnified image and quantification of immunohistochemical staining as positive tissue [pink, %] (**right graph**). N: Normoxemia (blue symbols); H: Hyperoxemia (red symbols); ipsilateral: blood-injected hemisphere (solid circles); contralateral: instrumented-only hemisphere (open circles).

**Table 1 ijms-25-06574-t001:** Primary antibodies.

Primary Antibody (Source, Catalog No., RRID)	Host Species	Immunogen Sequence	Concentration Used for IHC
**anti-Albumin** (Abcam, ab79960, AB_1658916)	Rabbit Polyclonal	Pig Albumin purified from pig plasma	1:2000
**anti-CBS** (Protein Tech, 14787-1-AP, AB_2070970)	Rabbit Polyclonal	CBS fusion protein Ag6437	1:200
**anti-CSE** (Protein Tech, 12217-1-AP, AB_2087497)	Rabbit Polyclonal	Gamma cystathionse fusion protein Ag2872	1:200
**anti-Nitrotyrosine** (Merck Millipore, ab5411, AB_177459)	Rabbit Polyclonal	Nitrated KLH	1:200
**anti-OT** (Millipore, Ab911, AB_2157629)	Rabbit Polyclonal	CYIQNCPLG (Synthetic oxytocin (Sigma) conjugated to thyroglobulin)	1:500
**anti-OT-R** (Protein Tech, 123045-1-AP, AB_2827425)	Rabbit Polyclonal	Oxytocin Receptor fusion protein Ag19074	1:100

## Data Availability

The data presented in this study are available on request from the corresponding author.

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
