# Peer review of "The Effect of Targeted Hyperoxemia on Brain Immunohistochemistry after Long-Term, Resuscitated Porcine Acute Subdural Hematoma and Hemorrhagic Shock"

_ijms, 2024, doi:10.3390/ijms25126574_

Round 1

Reviewer 1 Report

Comments and Suggestions for Authors

The manuscript entitled "The effect of targeted hyperoxemia on brain immunohisto- 2 chemistry after long-term, resuscitated porcine acute subdural 3 hematoma plus hemorrhagic shock" is original, but there are different major concerns to be addressed:

1) Quality of figure is really low, for my opinion I don't see the signal of the immunohistochemistry. The authors MUST improve the quality of the figures,

2) It is not possible to distinguish the different markers that they used in the experiments.

3) The authors should also furnish all negative controls (that they used to test the antibodies)

Reviewer 2 Report

Comments and Suggestions for Authors

I appreciate the authors for presenting this clinically useful research article that emphasizes the effects of hyperoxemia on SDH + HS. I agree with the authors' conclusion that targeted hyperoxemia during resuscitation from combined ASDH and HS did not exhibit any apparent adverse effects. My comments are as follows:

Materials and Methods

  1. Lines 297-298: "Microdialysis catheters and multimodal probes for brain monitoring were introduced bilaterally into the cerebral parenchyma." Why did you introduce microdialysis? Do you mean ICP was introduced bilaterally into the parenchyma?
  2. When was the blood gas sampling for PaO2 evaluation conducted?
  3. The authors mentioned, "The results are reported as the percentage of the positively stained area as a fraction of the total area." How many samples (slices) were calculated?
  4. There is a lack of dose-response effects of acute SDH (blood volume injection) on brain parameters.

Results

  1. Please provide details on the damaged region after the SDH injection. Was the prefrontal cortex involved?
  2. Please provide the MAP, ICP, and CPP data.
  3. What was the mortality rate in normoxemia and hyperoxemia?
  4. I would like to see the experimental protocol presented as a graphical time-flow diagram.

Discussion

Well discussed.

Round 2

Reviewer 1 Report

Comments and Suggestions for Authors

The authors improved the quality of the manuscript. 

Reviewer 2 Report

Comments and Suggestions for Authors

The revised manuscript has replied my comments item-by-item. I have no  more comments.